# Enzyme-mimetic self-catalyzed polymerization of polypeptide helices

Ziyuan Song [1,10], Hailin Fu [2,3,10], Ryan Baumgartner [4,10], Lingyang Zhu [5], Kuo-Chih Shih [3], Yingchun Xia [1], Xuetao Zheng [4], Lichen Yin [6], Christophe Chipot [7,8,9*], Yao Lin [2,3*] & Jianjun Cheng [1,4*]

Enzymes provide optimal three-dimensional structures for substrate binding and the subsequent accelerated reaction. Such folding-dependent catalytic behaviors, however, are seldom mechanistically explored with reduced structural complexity. Here, we demonstrate that the α-helix, a much simpler structural motif of enzyme, can facilitate its own growth through the self-catalyzed polymerization of N-carboxyanhydride (NCA) in dichloromethane. The reversible binding between the N terminus of α-helical polypeptides and NCAs promotes rate acceleration of the subsequent ring-opening reaction. A two-stage, Michaelis–Menten-type kinetic model is proposed by considering the binding and reaction between the propagating helical chains and the monomers, and is successfully utilized to predict the molecular weights and molecular-weight distributions of the resulting polymers. This work elucidates the mechanism of helix-induced, enzyme-mimetic catalysis, emphasizes the importance of solvent choice in the discovery of new reaction type, and provides a route for rapid production of well-defined synthetic polypeptides by taking advantage of self-accelerated ring-opening polymerizations.

[1] Department of Materials Science and Engineering, University of Illinois at Urbana-Champaign, Urbana, IL 61801, USA. [2] Department of Chemistry, University of Connecticut, Storrs, CT 06269, USA. [3] Polymer Program, Institute of Materials Science, University of Connecticut, Storrs, CT 06269, USA. [4] Department of Chemistry, University of Illinois at Urbana-Champaign, Urbana, IL 61801, USA. [5] NMR laboratory, School of Chemical Sciences, University of Illinois at Urbana-Champaign, Urbana, IL 61801, USA. [6] Institute of Functional Nano & Soft Materials (FUNSOM), Jiangsu Key Laboratory for Carbon-Based Functional Materials & Devices, Collaborative Innovation Center of Suzhou Nano Science & Technology, Soochow University, Suzhou 215123, China. [7] Theoretical and Computational Biophysics Group, Beckman Institute for Advanced Science and Technology, University of Illinois at Urbana-Champaign, Urbana, IL 61801, USA. [8] Department of Physics, University of Illinois at Urbana-Champaign, Urbana, IL 61801, USA. [9] Laboratoire International Associé Centre National de la Recherche Scientifique et University of Illinois at Urbana-Champaign, Unité Mixte de Recherche n° 7019, Université de Lorraine, B.P. 70239, 54506 Vandœuvre-lès-Nancy, cedex, France. [10] These authors contributed equally: Ziyuan Song, Hailin Fu, Ryan Baumgartner. *email: chipot@ks.uiuc.edu; yao.lin@uconn.edu; jianjunc@illinois.edu

Enzymes, folded from long chains of polypeptides, catalyze biochemical reactions that are otherwise slow under physiological conditions[1–4]. In a typical enzymatic catalysis, several spatially positioned amino-acid residues of enzymes coordinate the binding with the substrate[5], which facilitates the formation of the enzyme–substrate complex that stabilizes the transition states and lowers the activation energy of the reaction. Many enzymatic kinetics can be described by a simple, two-step model proposed by Michaelis and Menten[6], involving a reversible substrate binding and a subsequent reaction at an accelerated rate. The binding of the substrate and the formation of enzyme–substrate complex is usually a prerequisite to the catalytic reaction.

The folded, three-dimensional structure of enzyme is crucial for its binding with the substrate, which ensures the spatial preorganization of the binding sites for synergistic supramolecular interaction[1,2]. The mutation of amino acids in the catalytic region (e.g., cleft or pocket) or the mild denaturation of the enzymes can lead to significant loss of the enzyme activity[7,8]. An interesting question is whether a basic segment or a simpler molecular structure of enzyme, with reduced chemical and topological complexity, would elicit a Michaelis–Menten type of catalysis, capable of spatial- and site-specific substrate binding and accelerated reaction. We hypothesized that the α-helix, the fundamental structural motif in enzymes, would be such molecular structure, as its pre-aligned N–H groups at the N terminus, as well as C=O groups at the C terminus, may function as the binding sites with substrates containing complementary hydrogen bond (H-bond) moiety (Fig. 1a)[9]. Although the majority of the backbone N–H and C=O groups within an α-helical polypeptide are intramolecularly connected via H-bonds, the four C=O groups at the C terminus and the four N–H groups at the N terminus remain unbound owing to the lack of H-bonding partners (see Fig. 1 for details). These dangling bonds have a reduced degree of conformational freedom because of the helix H-bonding framework, and to some extent align with the direction of the macrodipole of the helix. Therefore, if a chemical reaction involves a substrate with H-bond donating or accepting ability, Michaelis–Menten type of catalysis can potentially be realized in the chain ends of a molecular structure as simple as an α-helix.

Figure 1b shows N-carboxyanhydride (NCA), an active monomer for the growth of polypeptides, that has three H-bond acceptors and one H-bond donor on its ring. In solvents with low polarity, such as dichloromethane (DCM) or chloroform, NCA may bind to the terminus of an α-helical polypeptide with unsaturated groups, including the N terminus that also functions as a nucleophile that opens the NCA ring and promotes the chain propagation (Fig. 1c). In this study, we report the self-catalyzed growth of α-helical polypeptide in a Michaelis–Menten manner. The binding of NCA with the N terminus is convincingly demonstrated through experimental and simulation-based methods, which promotes the polymerization in an accelerated rate. The fast polymerization kinetics allows us to synthesize well-defined polypeptide materials even in the presence of water, an otherwise challenging task for conventional NCA polymerization owing to water-induced side reactions.

## Results and discussion
**Linear polymerization of NCA in DCM with an accelerated rate.** Conventional NCA polymerization is conducted in solvents like N,N-dimethylformamide (DMF), which usually takes days to finish. During the elongated polymerization time, chain terminations are usually observed[10–12]. The synthesis of linear homopolypeptides from polymerization of NCAs in DCM (Fig. 2a), however, is found to be much faster. DCM, as a solvent of low dielectric constant, promotes H-bonding interactions between the helical polypeptides and the NCA molecules, to some extent simulating the specific, non-polar environment in the catalytic pocket in enzyme. For example, the polymerization of γ-benzyl-L-glutamate NCA (BLG-NCA) initiated by n-hexylamine in DCM revealed a two-stage kinetics (Fig. 2b and Supplementary Fig. 1)[10,13,14]. The second stage started when the growing

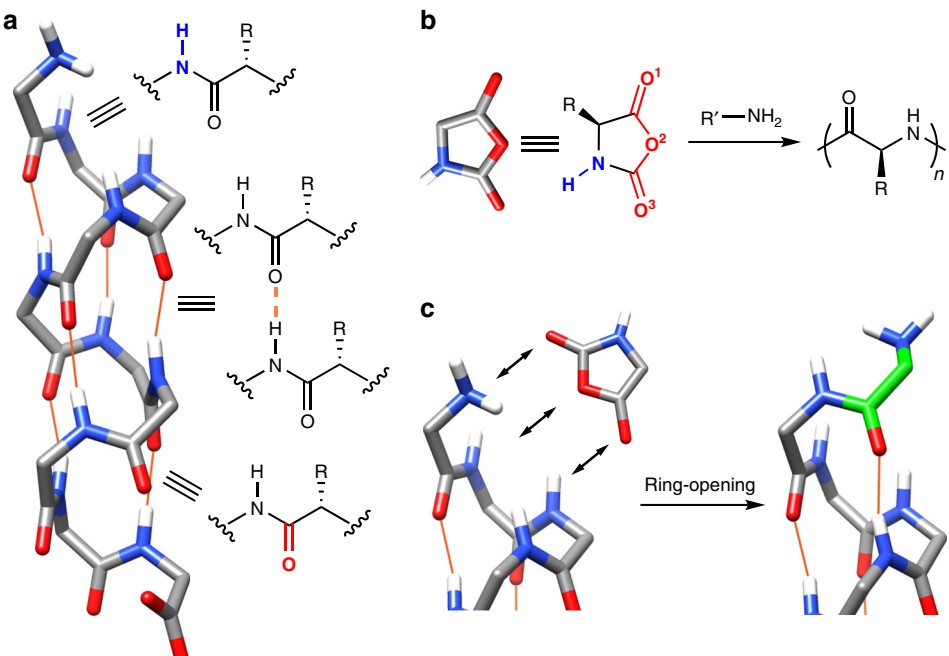

**Fig. 1** Schematic illustration of the catalytic power of an α-helix. **a** Illustration of possible binding sites at the termini of a free α-helical polypeptide. Intrahelical H-bonds are represented with orange lines. Side chains are omitted for simplicity. **b** Analysis of H-bond acceptors (red) and donors (blue) in an NCA molecule and its polymerization to synthesize polypeptides. **c** Proposed binding of NCA with the N terminus of an α-helix and subsequent ring-opening reaction for the growth of polypeptide. The newly formed residue from NCA is highlighted in green.

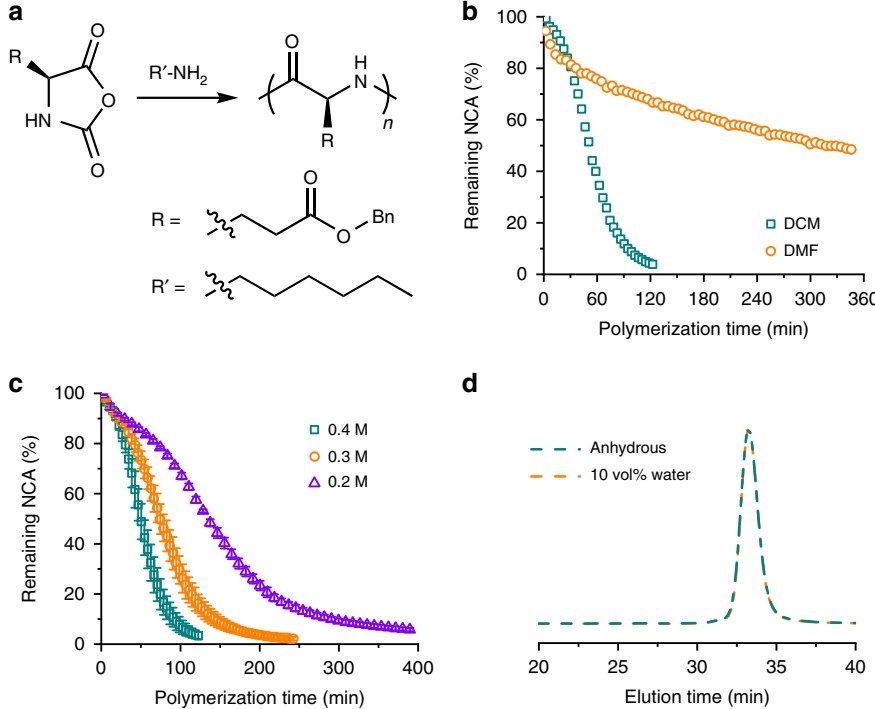

**Fig. 2** Polymerization kinetics of BLG-NCA. **a** Scheme showing the polymerization of BLG-NCA initiated by *n*-hexylamine. **b** Conversion of BLG-NCA in anhydrous DCM and DMF. $[M]_0 = 0.4$ M, $[M]_0/[I]_0 = 100$. **c** Conversion of BLG-NCA in DCM at various $[M]_0$. $[M]_0/[I]_0 = 100$. Error bars represent standard deviations from three independent measurements. **d** Normalized GPC-LS trace of obtained PBLG from polymerization of BLG-NCA in DCM under anhydrous conditions or in the presence of 10 vol% water. $[M]_0 = 0.4$ M, $[M]_0/[I]_0 = 100$.

polypeptide reached a degree of polymerization (DP) of ~ 10[15,16], with an apparent rate constant ~20 times larger than that of the first stage. The in situ Fourier-transform infrared spectroscopy characterization of polymerization in DCM exhibited the signals at 1649 and 1655 $cm^{-1}$ (Supplementary Fig. 1), indicating the formation of a coiled conformation in early stages of the polymerization. This result contrasts with the polymerization in dioxane[17], where β-sheet was observed at low $[M]_0/[I]_0$. Owing to the accelerated kinetics in the second stage, polymerization in DCM completed much faster than a conventional NCA polymerization in DMF. For instance, polymerization of BLG-NCA in DCM at $[M]_0 = 0.4$ M with $[M]_0/[I]_0 = 100$ reached 98% conversion after 2 h. In comparison, only 33% NCA was consumed in DMF under identical conditions (Fig. 2b). The polymerization rate in DCM increased significantly at higher $[M]_0$ (Fig. 2c and Supplementary Fig. 1), in sharp contrast with the polymerization in DMF, for which the reaction rate was almost independent of $[M]_0$ (Supplementary Fig. 2). The polypeptides synthesized in DCM showed predictable MWs and low dispersity ($Đ = M_w/M_n$, <1.25) (Table 1 and Supplementary Fig. 3), indicating well-controlled polymerization process.

The accelerated polymerization kinetics in DCM help outpace side reactions during NCA polymerization[18]. After the 2-h polymerization in DCM, the end-group analysis showed <1% loss of terminal amine groups[19], indicating negligible chain termination[10,12]. In addition, owing to the fast polymerization kinetics and the water-immiscibility of DCM[20,21], the polymerization of NCA in DCM in the presence of 10 vol% water still resulted in well-defined polypeptide with predictable MW, similar to that obtained under anhydrous conditions (Fig. 2d and Supplementary Table 1). In contrast, the presence of water in DMF as a solvent led to a 10% decrease in MW of the polypeptides due to degradation of NCA monomers in the slow polymerization process (Supplementary Fig. 2 and Supplementary Table 1).

**Table 1 Characterization of resulting polypeptides from linear polymerization of BLG-NCAs in DCM[a].**

| $[M]_0$ (M) | $[M]_0/[I]_0$ | $M_n/M_n{*}^{b,c}$ (kDa) | $Đ^c$ |
|---|---|---|---|
| 0.2 | 50 | 11.4/11.1 | 1.25 |
| | 100 | 23.9/22.0 | 1.09 |
| | 150 | 28.9/33.0 | 1.10 |
| 0.3 | 50 | 11.7/11.1 | 1.22 |
| | 100 | 23.2/22.0 | 1.07 |
| | 150 | 31.4/33.0 | 1.07 |
| 0.4 | 50 | 12.2/11.1 | 1.16 |
| | 100 | 23.5/22.0 | 1.07 |
| | 150 | 32.8/33.0 | 1.06 |

[a]All polymerizations were conducted in anhydrous DCM at room temperature initiated by *n*-hexylamine
[b]Obtained MW/designed MW*
[c]Determined by GPC

**Impact of solvents and NCA structures**. The solvent-dependent kinetics profile was further confirmed by running the polymerization of BLG-NCA in additional solvents with low dielectric constants, including chloroform ($ε = 4.81$) and 1,2-dichloroethane ($ε = 10.65$). Both polymerization exhibited two-stage, self-catalysis feature similar with that in DCM, reaching >98% conversion within 150 min (Supplementary Fig. 4). Furthermore, addition of toluene, a non-polar solvent with a low dielectric constant ($ε = 2.38$), further increases the polymerization rate in DCM, completing polymerization within 1 h at $[M]_0/[I]_0 = 50$ with $[M]_0 = 0.2$ M (Supplementary Fig. 4). Put together, these results confirm the universal self-catalysis feature of polymerization of NCA in solvents with low dielectric constants.

The self-catalytic, accelerated polymerizations in DCM were also observed for other NCAs with different chirality and side

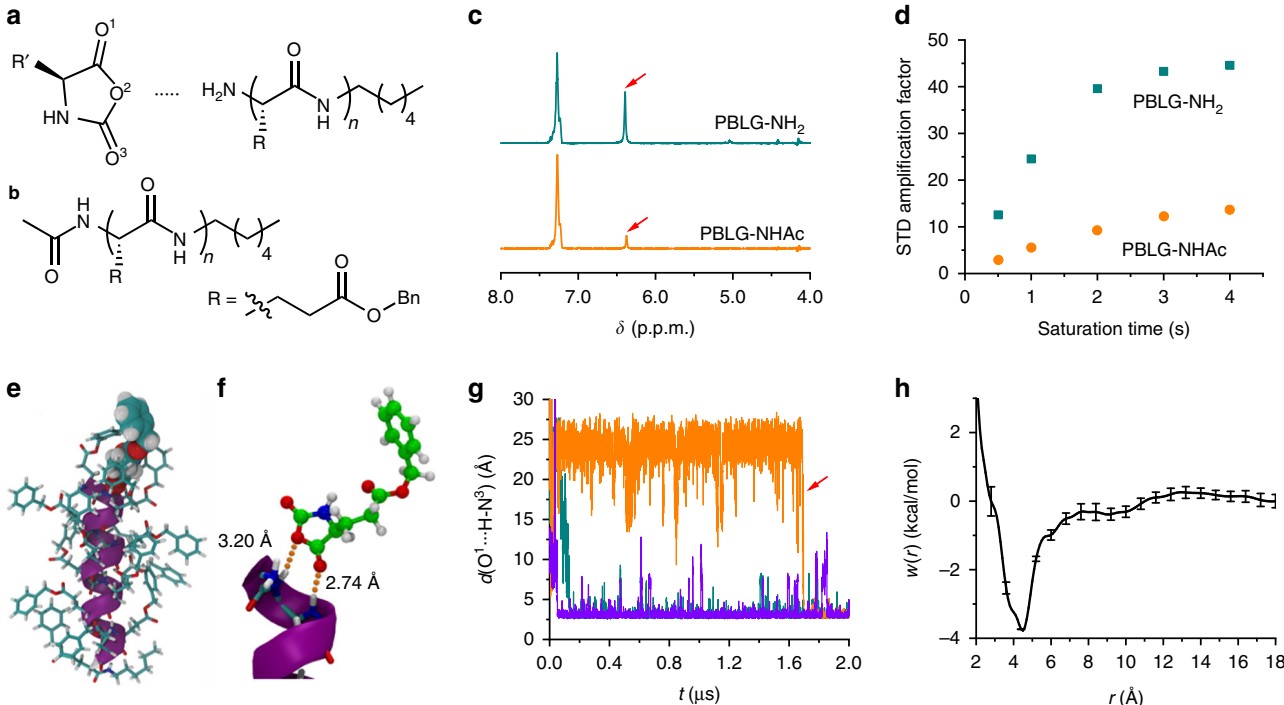

**Fig. 3** Reversible binding of polypeptide and NCA monomer. **a**, **b** Chemical structures of BLG-NCA, PBLG-NH$_2$ **a** and PBLG-NHAc **b**. **c** STD NMR spectra of ELG-NCA in the presence of PBLG-NH$_2$ or PBLG-NHAc. The STD signal of ring N–H proton is highlighted with red arrows. The peak at 7.27 ppm is the irradiation peak since no background suppression was applied. **d** STD amplification factor of ring N–H proton of ELG-NCA in the presence of PBLG-NH$_2$ or PBLG-NHAc at various saturation times. **e** Snapshot of simulation trajectories showing the binding between PBLG-NH$_2$ and BLG-NCA. The α-helical backbone is represented as a purple ribbon, the side chains of PBLG as sticks, and NCA as van der Waals spheres. **f** Closeup from the simulation trajectory at the N terminus of PBLG-NH$_2$ to reveal the H-bonding interactions (highlighted with the orange dotted lines). Side chains are omitted for simplicity. **g** Time evolution of the H bonds formed between BLG-NCA (O$^1$) and PBLG-NH$_2$ (amide N–H from third residue). Each line represents one independent, 2-μs long MD simulation. The red arrow indicates the transfer of NCA from C terminus to N terminus in the third simulation run. **h** PMF profile for the reversible binding of PBLG-NH$_2$ and BLG-NCA. Error bars correspond to estimated standard deviations from four independent walkers of the eABF algorithm.

chain structures. For instance, the polymerization of γ-benzyl-D-glutamate NCA (BDG-NCA) and $N^{\epsilon}$-2-(2-(2-methoxyethoxy)ethoxy)acetyl-L-lysine NCA (EG$_2$-Lys-NCA) in DCM also exhibited two-stage kinetics (Supplementary Fig. 5). On the other hand, it has to be noted that the chirality match between propagating polypeptides and NCA monomer is important for the rate acceleration. The polymerization of BLG-NCA initiated by poly(γ-benzyl-D-glutamate) (PBDG) macroinitiators, for instance, was slower initially compared with that initiated by PBLG macroinitiators (Supplementary Fig. 5), likely due to the switch of helical sense after the growth of several PBLG units on PBDG[10].

**Binding interaction of polypeptide and NCA**. The remarkable solvent-dependent and [M]$_0$-dependent polymerization behavior provided an impetus for us to elucidate the rate acceleration mechanism of polymerization of NCAs in DCM. Inspired by enzymatic catalysis modelled by Michaelis–Menten mechanism, we reasoned that there may exist a reversible, non-covalent binding of α-helical polypeptide propagating chains and NCA monomers, which stabilizes the transition states and lowers the activation energy for the ring-opening of NCAs. To test our hypothesis, STD NMR, a useful technique for probing protein–ligand interactions[22,23], was conducted for a mixture of amine-terminated, helical poly(γ-benzyl-L-glutamate) (PBLG-NH$_2$) and γ-ethyl-L-glutamate NCA (ELG-NCA) in CD$_2$Cl$_2$ (Fig. 3a). When PBLG-specific aromatic protons (7.27 ppm) were saturated, a strong STD signal arising from the ring N–H protons of ELG-NCA was observed (Fig. 3c), thereby suggesting binding

interactions of PBLG-NH$_2$ and ELG-NCA. In contrast, the STD signal was much weaker in DMF under similar conditions (Supplementary Fig. 6). The large dielectric constant ($\varepsilon = 36.7$) and high polarity of DMF solvent disrupt the binding interactions of the polypeptide and NCA, and contribute to a slower polymerization rate. To identify the binding site in the polypeptide, STD NMR was performed for acetyl-capped polypeptides (PBLG-NHAc, Fig. 3b and Supplementary Fig. 7) and ELG-NCA, which exhibited a decreased STD signal from the NCA ring N–H protons, indicative of weaker polypeptide-NCA binding (Fig. 3c). Further quantification revealed an 80% decrease of STD amplification factors upon acetyl-capping of PBLG-NH$_2$ at all saturation times (Fig. 3d), suggesting that the N terminus of PBLG serves as the primary active site for reversible NCA binding. In addition, binding interactions between polypeptides and NCA at the ring N–H protons were also observed from the STD NMR spectra of BDG-NCA/poly(γ-ethyl-L-glutamate) and EG$_2$-Lys-NCA/PBLG (Supplementary Fig. 5).

**Molecular dynamics simulations on the binding interactions**. Binding of polypeptides and NCAs was further examined by molecular dynamics (MD) simulations (Fig. 3e, f). As can be seen in Fig. 3g and Supplementary Fig. 8, binding of NCA and the N terminus of helical PBLG-NH$_2$ is consistently observed in all three independent, 2-μs equilibrium simulations, albeit over different timescales (see Supplementary Movies 1, 2 for details). Whereas BLG-NCA anchored to the N terminus within 0.2 μs in two simulations, association to the C terminus at first was observed in the third one, the BLG-NCA stayed at the C terminus

for some time, and moved to N terminus after ~1.7 µs (Fig. 3g). It is noteworthy that once formed, the complex is particularly robust, exhibiting no trend towards spontaneous dissociation. Comparison of the three trajectories revealed a common pattern in the association mechanism, namely the shortest H-bond is that established between $O^1$ of NCA and the third-residue nitrogen atom. In this relative orientation of the BLG-NCA, the H-bond formed between $O^2$ and the first-residue nitrogen atom is also the shortest (Fig. 3f).

The potential of mean force (PMF) underlying reversible association of BLG-NCA to the helical polypeptide exhibits a single minimum, 3.8 kcal/mol deep, at ~4.5 Å, corresponding to the bound state (Fig. 3h). In a typical binding pose, in addition to the network of H bonds established between the carbonyl moiety of NCAs and the N terminus of the α-helix, the complex is further stabilized by the formation of an attractive quadrupole–quadrupole, T-shaped π–π interaction[24] of NCA and the side chain of the last residue in PBLG (Supplementary Fig. 9). Integration of the PMF yields a standard binding free energy of ~−2.4 kcal/mol[25].

**The key role of H-bonding interactions for rate acceleration.** The folded α-helical conformation of a polypeptide has a critical role in its binding interaction with NCA. A closer look at the simulation trajectories revealed that the initial binding occurred only at the C terminus or N terminus, which bears unsaturated C=O or N–H groups for H-bonding interactions. The binding at C terminus is less stable, presumably owing to the formation of a single H-bond (only one ring N–H donor on the NCA ring) and the steric hindrance of hexyl group at the C terminus. The binding of NCAs with the reaction site, i.e., the N terminus of PBLG, is, therefore, facilitated owing to reduced non-specific bindings. In comparison, in a pentapeptide, $PBLG_5$, which adopts a coiled conformation with the polar amide groups exposed in DCM, a variety of NCA binding motifs were observed (Supplementary Fig. 10). These non-specific interactions interfere with the desired binding at the N terminus, which may lead to slower polymerization rates.

In order to demonstrate the importance of the dangling N–H groups at the N terminus of polypeptide for catalysis, we carried out the polymerization of sarcosine NCA initiated with an α-helical $PBLG-NH_2$ macroinitiator. Although the initial polymerization rate was comparable with the polymerization of BLG-NCA under identical conditions, the consumption of sarcosine NCA significantly slowed down as the polymerization proceeded (Supplementary Fig. 11), which was attributed to the absence of N–H bonds at the propagating terminus of poly(sarcosine). The $N-CH_3$ groups on poly(sarcosine) cannot provide H-bond donors

for the binding with sarcosine NCA monomers, leading to the decreased polymerization rate.

Moreover, the addition of 10 vol% dimethyl sulfoxide (DMSO), a strong H-bonding acceptor[26], significantly slowed down the polymerization of BLG-NCA in DCM, requiring ~250 min to reach 80% NCA conversion at $[M]_0/[I]_0 = 100$ with $[M]_0 = 0.4$ M (Supplementary Fig. 11), much longer than that in the absence of DMSO (~80 min). The increase in DMSO content to 50 vol% resulted in further decrease of the polymerization rate. In addition, the sigmoidal, two-stage kinetics profile disappeared upon addition of DMSO, suggesting the important role H-bonding plays in the accelerated polymerization.

**Two-stage kinetic model with an adsorption step.** With strong evidence of a reversible binding between α-helical polypeptides and NCAs, we established a two-stage kinetic model[16,27–29] with NCA adsorption step incorporated, which is described by the following successive reactions:

$$M_i^* + M \xrightarrow{k_1} M_{i+1}^* \qquad\qquad 1 \le i < s$$

$$M_i^* + M \underset{k_{off}}{\overset{k_{on}}{\rightleftarrows}} M_i^* - M \xrightarrow{k_r} M_{i+1}^* \quad i \ge s$$

where $M$ represents NCA monomer and $M_i^\star$ denotes a polymer chain with DP of $i$ and an active site (*) in the end. In the initial stage of the chain growth ($i < s$), we treat the reaction between the monomer and the active end of the coil chain as a second-order reaction with a rate constant $k_1$. When the growing chain reaches the critical length $s$ and folds into an α-helix, we consider the reaction to occur in two steps: first, the monomer binds to the active helical chain to form the reaction complex $M_i^\star - M$, with an adsorption rate constant $k_{on}$ and a desorption rate constant $k_{off}$; subsequently, the attack of the active end of the helical chain on the bound monomer triggers a ring-opening reaction and allows the chain elongation with a rate constant $k_r$. Based on this model, it is then standard practice to write the kinetic equations corresponding to the above scheme (see Supplementary Methods), and determine their numerical solutions.

Figure 4a shows the predicted kinetic curves from solving the differential equations numerically for an identical set of $s$, kinetic rate constants, and $[M]_0/[I]_0$, but with a different $[M]_0$. The four kinetic curves in the dimensionless time form ($\tau = tk_1[M]_0$) differ from each other at the second stage; their shapes are a function of $[M]_0$ (Supplementary Fig. 12). The molecular-weight distribution (MWD) of the polymers predicted from the parameters of the kinetic model is slightly asymmetric (Fig. 4b), deviating from the Poisson distribution from a typical chain growth mechanism[30]. In addition, some very short chains (DP < s) may still be found in

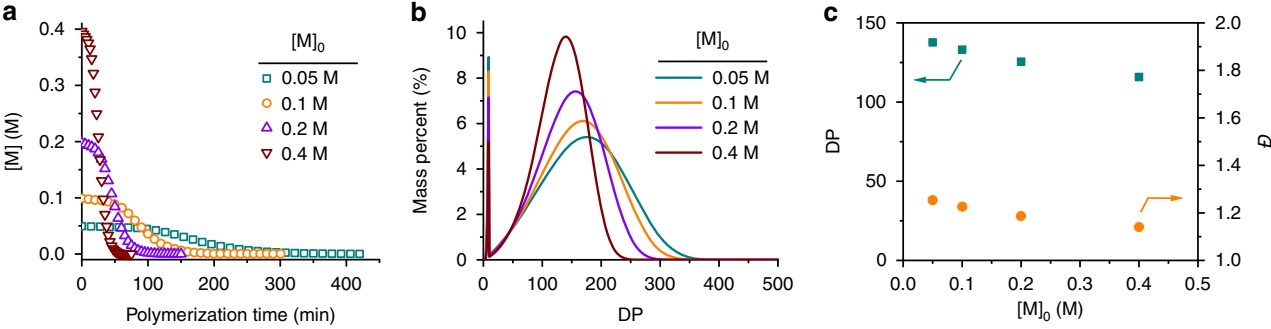

**Fig. 4** Simulations with adsorption-incorporated, two-stage kinetic model. **a** Plot of the monomer concentration vs. time for test cases with $s = 10$, $[M]_0/[I]_0 = 100$, $k_1 = 0.02$ $M^{-1}s^{-1}$, $k_{on} = 10$ $M^{-1}s^{-1}$, $k_{off} = 2$ $s^{-1}$, $k_r = 0.2$ $s^{-1}$, and at selected values of $[M]_0 = 0.05$, 0.1, 0.2, or 0.4 M. **b** Predicted MWD profiles based on the kinetic profiles in **a**. **c** Calculated DP and Đ at various $[M]_0$.

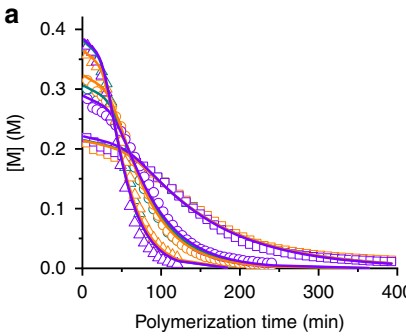
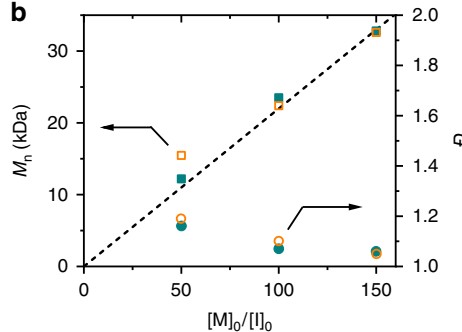

**Fig. 5** Kinetic modeling of polymerization of NCA in DCM. **a** Polymerization kinetics in Fig. 2c were fit with the adsorption-incorporated kinetic model. **b** Comparison of experimental results obtained from GPC analysis (solid symbols) and predicted results from modeling (open symbols) with various $[M]_0/$$[I]_0$ ratios at $[M]_0 = 0.4$ M. The dashed line represents designed MW by $[M]_0/[I]_0$ ratio.

the end of the reaction, their mass percentages being a function of the initial monomer concentration. The dependence of the DP and Đ on $[M]_0$ is plotted in Fig. 4c. Because of the asymmetric nature of the MWD, the relatively small variation in the calculated Đ may not fully account for the difference in the MWD in Fig. 4b. Overall, the substantial dependence of the kinetic curves and MWDs on the initial concentration of the monomers is a key characteristic of this kinetic model.

**Fitting of polymerization kinetics with the model**. The kinetic model was subsequently utilized to fit the polymerization of NCAs in DCM. By sharing the same $k_r$ and $s$, but allowing $k_1$ and $k_{on}/k_{off}$ to be individually optimized, a global fit was successfully obtained to nine sets of kinetic data with varied $[M]_0$ and $[I]_0$ (Fig. 5a, Supplementary Fig. 13 and Supplementary Table 2). Likely, the smaller binding free energy obtained from the kinetic modeling compared with the simulation-based calculation is attributed to the strong aggregation tendency of NCAs in DCM, which interferes with the reversible binding between polypeptide and NCA. In addition, possible force-field inaccuracies cannot be discounted. Nonetheless, the model-predicted MWs and MWDs based on the estimated rate constants agree well with the GPC-LS traces of the resulting polymers (Fig. 5b and Supplementary Figs. 14, 15), suggesting that the two-stage kinetic model with a Michaelis–Menten feature successfully describe the polymerization of NCAs in DCM.

In conclusion, we demonstrate that the helix-induced, self-catalyzed polymerization of NCAs shows Michaelis–Menten type, enzyme-mimetic characteristics in selected solvents with low dielectric constants. The binding between the growing helical chains and the incoming monomers preceding to the reaction is a crucial step in this unique polymerization. The use of DCM as a solvent obviates the necessity to prevent polar, catalytic groups from interacting with polar solvents like water and DMF, thus, offering a unique system to reveal the catalytic power of a single helix. We believe that the present work will not only contribute to improving our current understanding of biological catalysis, but also offer a simplified, practical way to synthesize well-defined polypeptides.

## Methods

**Polymerization kinetics**. In a glovebox, γ-benzyl-L-glutamate NCA (BLG-NCA) and *n*-hexylamine were dissolved in $CD_2Cl_2$. The resulting solutions were mixed at pre-determined ratios, transferred into an NMR tube, and the NMR spectrum was monitored at different time intervals. The conversion of NCA monomers over time was then calculated according to the integrations in NMR spectra. The polymerization kinetics in other solvents (e.g., DMF, chloroform) and the polymerization kinetics of other monomers (e.g., BDG-NCA and sarcosine NCA) were conducted in a similar way.

**Saturation transfer difference nuclear magnetic resonance (STD NMR)**. Polypeptide was mixed with NCA in 1:100 ratio in $CD_2Cl_2$ and transferred into an NMR tube in a glovebox. STD NMR was performed at 25 °C with 128 scans at various saturation times, with the polypeptide-specific signals saturated.

**Molecular dynamics simulation**. All the molecular dynamics simulations were performed using NAMD 2.12[31], using the CHARMM36 force field[32] for the polypeptide chain and the CHARMM generalized force field[33] for the BLG-NCA and DCM. Three independent, $2 \times 10^{-6}$ s equilibrium simulations were performed, prefaced by a $2 \times 10^{-9}$ s thermalization. The extended-Lagrangian adaptive biasing force algorithm[34,35] was used to determine the PMF, $w(r)$, through integration of the average force exerted along the transition coordinate, $r$.

**Kinetic modeling**. The differential equations (see Supplementary Methods) were solved numerically using ode15s in Matlab. The rate constants were obtained by minimizing the sum of squared error between the simulated results and the experimental data. As for the fitting of the nine groups of experimental results ($[M]_0 = 0.2, 0.3, 0.4$ M, $[M]_0/[I]_0 = 50, 100, 150$), we set $k_r$ to be a global factor and let $k_1$ and $k_{on}/k_{off}$ to be individual values for the nine different conditions.

## Data availability

The data that support the findings of this study are available within the paper and its Supplementary Information files. Any other data are available from the corresponding authors upon reasonable request. The source data underlying Fig. 2c, Supplementary Figs. 1a, b are provided as a Source Data file.

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

## Acknowledgements

J.C. and Y.L. acknowledge funding from National Science Foundation (CHE-1709820 to J.C. and DMR-1809497 to Y.L.). C.C. acknowledges funding from the Contrat Plan État-Région Santé ITM2P and the European Regional Development Fund.

## Author contributions

Z.S., H.F., R.B., C.C., Y.L., and J.C. conceived and designed the experiments. Z.S., H.F., R.B., L.Z., K.-C.S., Y.X., X.Z., L.Y., and C.C. performed the experiments. Z.S., H.F., C.C., Y.L., and J.C. analyzed data and prepared the manuscript with contributions from all authors.

## Competing interests

The authors declare no competing interests.
