## [Transparent Peer Review File · Nature Communications]

Reviewers' comments:

Reviewer #1 (Remarks to the Author):

ROP with Enzyme-like kinetic features were described in this paper. The authors employed experimental, theoretical modeling, and simulation methods collectively and convincingly show that an initial binding process occurred between the helical polypeptide chain end and the incoming NCA monomer. This feature in turn rendered typical Michaelis–Menten kinetics. It was quite amazing to see how perfectly the experimental and theoretical numbers fit in terms of the kinetics, MW, and MWD. This is a conceptually new understanding to people working in NCA field and perhaps also more broader audiences. Therefore, I support its publication after some minor points are properly addressed:

1. it would be helpful if the authors can add some H-Bonding disruption/protein denature reagents and monitor the kinetics. This will further confirm the role of hydrogen bonding.
2. I expect the weak binding ($K_{eq} = 5 \text{ M}^{-1}$) between the NCA monomer and polypeptide would be sensitive to temperature. It would be interesting to do temperature dependent studies and see how well the experimental and theoretical data fits.
3. Monitoring the ROP kinetics using DL-NCA or N-substituted NCAs in DCM will serve as good control groups.
4. Does the macrodipole and/or helix-helix bundling affect the ROP in DCM at a relatively high $[M]_0$?
5. "While majority of the N–H and C=O groups within an α -helical polypeptide are intramolecularly connected via H-bonds, the four C=O groups at the C-terminus and the four N–H groups at the N-terminal remain unbound. These groups have reduced degree of conformational freedom because of the helix H-bonding framework, to some extent align along the direction of the macrodipole of the helix." This part reads confusing and it is not clearly to me which N-H and C=O are referred as "these groups".

Reviewer #2 (Remarks to the Author):

This manuscript reports a comprehensive study of self-catalyzed polymerization of N-carboxyanhydride (NCA) in dichloromethane (DCM). Compared with other NCA polymerization performed in more polar solvents, a reversible binding of NCA to the growing helical polypeptide promotes rate acceleration of the subsequent ring-opening reaction. It results that the corresponding kinetics shows Michaelis-Menten type, enzyme-mimetic characteristics in DCM. Overall, the work is comprehensively described and the manuscript may represent a 1) significant step further for the scientific community in the preparation and application of synthetic polypeptides polymers; 2) an intriguing study in which Michaelis-Menten-type kinetic model could be applied in polymer chemistry. All in all, this manuscript contains some interesting observations, but few more experimental works is necessary to better demonstrate how (or until where) the proposed self-catalyzed ROP is "superior" in practice to known ROP of NCA (see my comments below). Hence, I would recommend a publication of this manuscript in Nature Communication only after major corrections.

I have three main concerns regarding the work submitted by Jianjun Cheng and coworkers:

- 1) My first concern relies on the part dealing with the polymerization performed with an alternative solvent system containing 10% of water: the reference 17 cited by authors already presents how the use of an emulsion to achieve the ROP of BLG-NCA is original and relevant. Considering this reference 17, I do not really see where this emulsion system brings something original in this work and the corresponding paragraph could be summarized in only 1 or 2 sentences to improve the clarity (and the main message) of the manuscript. On the other hand, and concerning the solvent system, the overall work suggests that the specific kinetic behavior observed by the authors in

DCM is expected for solvents with low dielectric constant (see author's conclusions). Whereas I agree with this statement, this should be supported by few other kinetic runs in other solvent such as Toluene.

2) My second concern is related to the previous data published about the polymerization of BLG-NCA (see for instance M. Idelson and E. R. Blout, *J. Am. Chem. Soc.*, 1957, 79, 3948). At low polymerization degrees (Dp), PBLG adopts beta sheet conformations (below Dp 10), a feature that was omitted by the authors in this article. This, in part, makes somehow complicated the ROP of BLG-NCA in solvents such as DCM and is often at the origin of increased polydispersity upon the ROP, particularly when low Dps are targeted (as compared to DMF or THF, two other solvents that are generally used for the polymerization). Therefore:

- The two-stage kinetics presented by the authors should include this comment and it would be an important point to compare the affinity constant of the NCA in both cases (alpha helix and beta sheet).
- Regarding the pentapeptide PBLG5, either authors should present IR spectroscopy data supporting the coil conformation or they should revise the corresponding results and discussion.
- The polydispersity obtained for Dp50 from hexylamine in DCM should be compared to the polydispersity obtained from a DP20 prepared in DMF that is then used as macroinitiator to reach a Dp50 in DCM. Overall, it is to point out that the use of MALS detection might strongly minimize the signal of beta sheet oligomers and SEC traces with UV detection should also be given to support the SEC traces based on MALS detection.

3) My third concern is related to the impact of the two stages kinetics model proposed by the authors. I agree that such results may deserve a high-impact communication; nevertheless authors should better substantiate possible discrimination introduced by the proposed affinity towards helical polypeptides:

- is this affinity influenced by the chirality of the monomer (for instance, L-BLG-NCA versus D-BLG-NCA)? Several articles published by Paul Doty could indeed help the authors for discussion.
- is this affinity influenced by the side chain of the monomer (for instance, L-Lys-NCA versus L-BLG-NCA)? Few experiments to better substantiate these influences should be provided by the authors.

As a minor comment, authors should discuss why they do not observe pyroglutamate side reactions (intramolecular cyclization of the amino end-group with the adjacent benzyl ester) at room temperature in DCM (see the reference 12 given by authors). If ROP in DCM was performed at room temperature, was the MALDI-TOF presented as supplementary in figure 5 showing this pyroglutamate?

RESPONSES TO REFEREES (NCOMMS-19-20208):

(Reviewer comments in black, our response in blue, and text added to the paper in magenta)

Reviewer #1 (Remarks to the Author):

ROP with Enzyme-like kinetic features were described in this paper. The authors employed experimental, theoretical modeling, and simulation methods collectively and convincingly show that an initial binding process occurred between the helical polypeptide chain end and the incoming NCA monomer. This feature in turn rendered typical Michaelis–Menten kinetics. It was quite amazing to see how perfectly the experimental and theoretical numbers fit in terms of the kinetics, MW, and MWD. This is a conceptually new understanding to people working in NCA field and perhaps also more broader audiences.

We appreciate the positive comments of the Reviewer.

Therefore, I support its publication after some minor points are properly addressed:

1. It would be helpful if the authors can add some H-Bonding disruption/protein denature reagents and monitor the kinetics. This will further confirm the role of hydrogen bonding.

We are thankful for this suggestion. Since common H-bonding disruption reagents will either interfere with polymerization (*e.g.*, trifluoroacetic acid) or have poor solubility in chlorinated solvents (*e.g.*, urea), we selected dimethyl sulfoxide (DMSO) as a strong H-bonding acceptor (*J. Am. Chem. Soc.*, **1976**, 98, 377) to study its impact on polymerization kinetics. Addition of 10 vol% DMSO significantly slowed down the polymerization of NCA in DCM, with slower polymerization observed when the DMSO content increased to 50 vol%, which confirmed the important role H-bonding played in the accelerated polymerization of NCA in DCM. This new result has been included in the revised manuscript as Supplementary Fig. 11b (Page S19, highlighted). The following discussion has been added to the main text to describe this new experiment and highlight the role of H-bonding (Page 13, highlighted):

“Moreover, the addition of 10 vol% dimethyl sulfoxide (DMSO), a strong H-bonding acceptor, significantly slowed down the polymerization of BLG-NCA in DCM, requiring ~ 250 min to reach 80% NCA conversion at $[M]_0/[I]_0 = 100$ with $[M]_0 = 0.4$ M (Supplementary Fig. 11), much longer than that in the absence of DMSO (~ 80 min). The increase in DMSO content to 50 vol% resulted in further decrease of the polymerization rate. In addition, the sigmoidal, two-stage kinetics profile disappeared upon addition of DMSO, suggesting the important role H-bonding plays in the accelerated polymerization.”

2. I expect the weak binding ($K_{eq} = 5$ M⁻¹) between the NCA monomer and polypeptide would be sensitive to temperature. It would be interesting to do temperature dependent studies and see how well the experimental and theoretical data fits.

The Reviewer is correct on the sensitivity of the binding to temperature. In addition, the folding of the helical PBLGs is also sensitive to the temperature (*Biomacromolecules*, **2017**, 18, 2324).

We have tested a T-dependent model that integrates the kinetic model with the folding equilibrium, but it requires a very large data set to determine the parameters unambiguously. The quantitative analysis of the temperature effect will be the subject of future study.

3. Monitoring the ROP kinetics using DL-NCA or N-substituted NCAs in DCM will serve as good control groups.

We are thankful for this suggestion. We have carried out the polymerization of sarcosine NCA (*i.e.*, *N*-methyl glycine NCA) from an α -helical PBLG macroinitiator. Poly(sarcosine) has weak hydrogen bonding donating ability due to the block of N-H by the methyl group. The polymerization exhibited an initial fast consumption of sarcosine NCA monomer, followed by a decrease in polymerization rate, which further confirmed our hypothesis of helix-promoted catalysis. The initial fast polymerization rate was attributed to the binding between sarcosine NCA with PBLG at the *N*-terminus through hydrogen bonding, which disappeared as the polymerization proceeded due to the absence of dangling N-H bonds on the propagating poly(sarcosine). This data has been included in the revised manuscript as Supplementary Fig. 11a (Page S19, highlighted). The following discussion has been added discuss this control experiment (Page 12-13, highlighted):

“In order to demonstrate the importance of the “dangling” N-H groups at the *N*-terminus of polypeptide for catalysis, we carried out the polymerization of sarcosine NCA initiated with an α -helical PBLG-NH₂ macroinitiator. While the initial polymerization rate was comparable with the polymerization of BLG-NCA under identical conditions, the consumption of sarcosine NCA significantly slowed down as the polymerization proceeded (Supplementary Fig. 11), which was attributed to the absence of N-H bonds at the propagating terminus of poly(sarcosine). The N-CH₃ groups on poly(sarcosine) cannot provide H-bond donors for the binding with sarcosine-NCA monomers, leading to the decreased polymerization rate.”

4. Does the macrodipole and/or helix-helix bundling affect the ROP in DCM at a relatively high [M]₀?

The macrodipoles and/or helix-helix bundling, indeed, affect polymerization at high [M]₀, as we observed larger rate constants with an increase in [I]₀ (*i.e.*, higher concentration of polypeptide chains) (Supplementary Table 2). However, the detailed mechanism clearly falls beyond the scope of this paper. We, therefore, focused our discussion on the behavior of one NCA with a single polypeptide chain, and, for the sake of clarity of our main message, will relegate the elucidation of chain-chain interactions and the interactions with multiple NCA molecules to a follow-up publication.

5. “While majority of the N-H and C=O groups within an α -helical polypeptide are intramolecularly connected via H-bonds, the four C=O groups at the C-terminus and the four N-H groups at the N-terminal remain unbound. These groups have reduced degree of conformational freedom because of the helix H-bonding framework, to some extent align along the direction of the macrodipole of the helix.” This part reads confusing and it is not clearly to me which N-H and C=O are referred as “these groups”.

We thank the Reviewer for these suggestions. “These groups” were originally used to refer to the four C=O groups at the C-terminus and the four N–H groups at the N-terminus, which do not have H-bonding partners to stabilize. We have rephrased the sentences to clarify the discussion (Page 4-5, highlighted).

“While the majority of the backbone N–H and C=O groups within an α -helical polypeptide are intramolecularly connected via H-bonds, the four C=O groups at the C-terminus and the four N–H groups at the N-terminus remain unbound due to the lack of H-bonding partners (see Fig. 1 for details). These “dangling bonds” have a reduced degree of conformational freedom because of the helix H-bonding framework, and to some extent align with the direction of the macrodipole of the helix.”

Reviewer #2 (Remarks to the Author):

This manuscript reports a comprehensive study of self-catalyzed polymerization of N-carboxyanhydride (NCA) in dichloromethane (DCM). Compared with other NCA polymerization performed in more polar solvents, a reversible binding of NCA to the growing helical polypeptide promotes rate acceleration of the subsequent ring-opening reaction. It results that the corresponding kinetics shows Michaelis-Menten type, enzyme-mimetic characteristics in DCM. Overall, the work is comprehensively described and the manuscript may represent a 1) significant step further for the scientific community in the preparation and application of synthetic polypeptides polymers; 2) an intriguing study in which Michaelis-Menten-type kinetic model could be applied in polymer chemistry.

We thank the Reviewer for these positive comments.

All in all, this manuscript contains some interesting observations, but few more experimental works is necessary to better demonstrate how (or until where) the proposed self-catalyzed ROP is "superior" in practice to known ROP of NCA (see my comments below). Hence, I would recommend a publication of this manuscript in Nature Communication only after major corrections.

We are thankful for these comments. We performed additional experiments to study the impact of the solvent, NCA chirality, and NCA side chains (check the detailed responses below), which helps define the scope of the self-catalyzed ROP of NCAs in low-polarity solvents, compared with conventional ROP methods. Generally speaking, the enzyme-like, self-catalyzed polymerization has faster polymerization kinetics that elicits a fast and efficient synthesis of homopolypeptides with controlled MWs and narrow dispersity. The added value of accelerated polymerization is the suppression of side reactions, including the water-induced NCA degradations (as demonstrated in the original manuscript) and the reservation of end-group fidelity (check the last response).

I have three main concerns regarding the work submitted by Jianjun Cheng and coworkers:

1) My first concern relies on the part dealing with the polymerization performed with an alternative solvent system containing 10% of water: the reference 17 cited by authors already presents how the use of an emulsion to achieve the ROP of BLG-NCA is original and relevant. Considering this

reference 17, I do not really see where this emulsion system brings something original in this work and the corresponding paragraph could be summarized in only 1 or 2 sentences to improve the clarity (and the main message) of the manuscript.

We are thankful for this comment, and agree with the Reviewer that the ROP in the presence of water is not conceptually novel. We have shortened the corresponding discussions. On the other hand, the impact of solvent was not studied in our previous paper about the polymerization in the presence of water (*Proc. Natl. Acad. Sci. USA*, **2019**, *116*, 10658). As a comparison, the self-catalyzed polymerization reported in this article provides a comparison with the polymerization in DMF, where the presence of water significantly degrades NCA monomers during the elongated polymerization process. We, therefore, left two sentences in the chemical synthesis section to highlight this comparison.

On the other hand, and concerning the solvent system, the overall work suggests that the specific kinetic behavior observed by the authors in DCM is expected for solvents with low dielectric constant (see author's conclusions). Whereas I agree with this statement, this should be supported by few other kinetic runs in other solvent such as Toluene.

We thank the Reviewer for this comment. We have added additional results on polymerization in solvents with a low dielectric constant, including chloroform and 1,2-dichloroethane (Supplementary Fig. 4a, Page S12, highlighted). Both polymerizations exhibited two-stage, self-catalyzed kinetics. The results about the studies with other non-polar solvents are consistent with our observation for a brush-polymer system (*Nat. Chem.*, **2017**, *9*, 614). In addition, although we cannot directly carry out the polymerization in toluene, as the Reviewer suggested, due to poor solubility of NCA monomer in toluene, we were able to polymerize BLG-NCA in a toluene/DCM co-solvent (1:3, v/v), where even faster polymerization was observed due to the lower dielectric constant of toluene ($\epsilon = 2.38$), compared with that of DCM ($\epsilon = 9.08$) (Supplementary Fig. 4b, Page S12, highlighted). The following text has been included in the manuscript to discuss the impact of the solvent (Page 8, highlighted):

“The solvent-dependent kinetics profile was further confirmed by running the polymerization of BLG-NCA in additional solvents with low dielectric constants, including chloroform ($\epsilon = 4.81$) and 1,2-dichloroethane ($\epsilon = 10.65$). Both polymerization exhibited two-stage, self-catalysis feature similar with that in DCM, reaching > 98% conversion within 150 min (Supplementary Fig. 4). Furthermore, addition of toluene, a non-polar solvent with a low dielectric constant ($\epsilon = 2.38$), further increases the polymerization rate in DCM, completing polymerization within 1 h at $[M]_0/[I]_0 = 50$ with $[M]_0 = 0.2$ M (Supplementary Fig. 4). Put together, these results confirm the universal self-catalysis feature of ROP of NCA in solvents with low dielectric constants.”

2) My second concern is related to the previous data published about the polymerization of BLG-NCA (see for instance M. Idelson and E. R. Blout, *J. Am. Chem. Soc.*, 1957, *79*, 3948). At low polymerization degrees (D_p), PBLG adopts beta sheet conformations (below D_p 10), a feature that was omitted by the authors in this article. This, in part, makes somehow complicated the ROP of BLG-NCA in solvents such as DCM and is often at the origin of increased polydispersity upon the

ROP, particularly when low Dps are targeted (as compared to DMF or THF, two other solvents that are generally used for the polymerization).

We appreciate the insightful comment from the Reviewer. In fact, we did not observe any formation of β -sheet structures during the ROP of BLG-NCA in DCM through FTIR characterization at $[M]_0 = 0.05$ M (The release of CO_2 and the volatility of DCM solvent make it difficult for us to directly probe the *in situ* IR at high $[M]_0$ in a sealed IR cell). The increase in intensity at 1649 and 1655 cm^{-1} indicated the formation of a mixture of α -helices and random coils (amide I), and the absence of IR signals at ~ 1630 cm^{-1} clearly suggested negligible β -sheet structures. As the polymerization proceeded, the signal coalesced into one peak at 1653 cm^{-1} , consistent with the formation of an α -helical conformation. This new result has been added as Supplementary Fig. 1c (Page S9, highlighted) and supports our argument of a coil-to-helix transition during ROP of NCA in DCM. On the other hand, we have added a short paragraph into our main text to discuss the coil-to-helix transition in early stage polymerizations (Page 6, highlighted):

“The *in situ* FTIR characterization of polymerization in DCM exhibited the signals at 1649 and 1655 cm^{-1} (Supplementary Fig. 1), indicating the formation of a coiled conformation in early stages of the polymerization. This result contrasts with the polymerization in dioxane, where β -sheet was observed at low $[M]_0/[I]_0$ ”

Therefore:

- The two-stage kinetics presented by the authors should include this comment and it would be an important point to compare the affinity constant of the NCA in both cases (alpha helix and beta sheet).

We thank the Reviewer for this suggestion. While we are interested in exploring the affinity of NCAs with β -sheet polypeptides, we were not able to find a suitable, solely β -sheet polypeptide in DCM for the binding studies. Common β -sheet forming synthetic polypeptides, such as poly(L-valine) and poly(O-benzyl-L-serine), form precipitates (homopolypeptides) or organogels (PEG copolymers) in DCM.

- Regarding the pentapeptide PBLG5, either authors should present IR spectroscopy data supporting the coil conformation or they should revise the corresponding results and discussion.

We thank the Reviewer for this comment. The PBLG 5-mer experiment is not critical to our main story, so we follow the suggestion of the Reviewer to remove this result.

- The polydispersity obtained for Dp50 from hexylamine in DCM should be compared to the polydispersity obtained from a DP20 prepared in DMF that is then used as macroinitiator to reach a Dp50 in DCM. Overall, it is to point out that the use of MALS detection might strongly minimize the signal of beta sheet oligomers and SEC traces with UV detection should also be given to support the SEC traces based on MALS detection.

We thank the Reviewer for these comment. Since the β -sheet oligomers were not observed during polymerization of BLG-NCA in DCM, we feel that the two-step chain extension study may not

provide any new information on the polymerization process. Nevertheless, we have provided the GPC-UV trace of the obtained polymers here (Fig. R1) to demonstrate the negligible generation of low-MW oligomers for ROP of NCAs in DCM.

Fig. R1. Overlaid GPC-UV ($\lambda = 260$ nm) and GPC-LS results of resulting PBLG polypeptide from ROP of BLG-NCA in DCM ($[M]_0 = 0.4$ M, $[M]_0/[I]_0 = 100$). The GPC-UV trace suggest the negligible formation of oligomers.

3) My third concern is related to the impact of the two stages kinetics model proposed by the authors. I agree that such results may deserve a high-impact communication; nevertheless authors should better substantiate possible discrimination introduced by the proposed affinity towards helical polypeptides:

- is this affinity influenced by the chirality of the monomer (for instance, L-BLG-NCA versus D-BLG-NCA)? Several articles published by Paul Doty could indeed help the authors for discussion.

- is this affinity influenced by the side chain of the monomer (for instance, L-Lys-NCA versus L-BLG-NCA)? Few experiments to better substantiate these influences should be provided by the authors.

We are thankful for these comments. We have carried out the polymerization of γ -benzyl-D-glutamate NCA (BDG-NCA, different chirality) and N^{ϵ} -2-(2-(2-methoxyethoxy)ethoxy)acetyl-L-lysine NCA (EG₂-Lys-NCA, different side chain), both showed the self-catalyzed, two-stage kinetics. In addition, we have conducted additional STD-NMR experiments with BDG-NCA and EG₂-Lys-NCA. In both cases, binding of NCAs with polypeptides through ring N-H proton was observed. However, due to the difference in NCA purity between batches, NCA solubility, side-chain interactions (*e.g.*, H-bonding between the side chains of poly(L-lysine) derivatives), it is difficult to make direct and quantitative comparison of binding affinity between these new monomers with the BLG-NCA. In fact, gelation occurred during the polymerization of N^{ϵ} -carboxybenzyl-L-lysine NCA (ZLL-NCA) due to the side-chain H-bonding, leading to super-fast polymerization that completed within 20 min (Fig. R2).

Fig. R2. Conversion of ZLL-NCA in DCM ($[M]_0 = 0.2$ M, $[M]_0/[I]_0 = 50$). Gelation occurred during the polymerization that resulted in super-fast polymerization.

On the other hand, it has to be noted that the polymerization kinetics were dependent on the chirality match between propagating polypeptides and NCA monomers. The polymerization rate of BLG-NCA initiated by PBDG macroinitiators (left-handed, unmatched chirality) was slower than that initiated by PBLG macroinitiator (right-handed, matched chirality), which was attributed to the switch of helical sense during the chain extension of PBLG from PBDG. Similar results were also reported by Doty and co-workers (*J. Am. Chem. Soc.* **1957**, 79, 3961).

The polymerization kinetics and new STD-NMR experiments were included as Supplementary Fig. 5 (Page S13, highlighted). The following paragraphs were included in the main text to discuss the impact of monomers (chirality and side-chain structure) (Page 8-9 and 10, highlighted):

“The self-catalytic, accelerated polymerizations in DCM were also observed for other NCAs with different chirality and side chain structures. For instance, the polymerization of γ -benzyl-D-glutamate NCA (BDG-NCA) and *N*^c-2-(2-(2-methoxyethoxy)ethoxy)acetyl-L-lysine NCA (EG₂-Lys-NCA) in DCM also exhibited two-stage kinetics (Supplementary Fig. 5). On the other hand, it has to be noted that the chirality match between propagating polypeptides and NCA monomer is important for the rate acceleration. The polymerization of BLG-NCA initiated by poly(γ -benzyl-D-glutamate) (PBDG) macroinitiators, for instance, was slower initially compared with that initiated by PBLG macroinitiators (Supplementary Fig. 5), likely due to the switch of helical sense after the growth of several PBLG units on PBDG.”

“Additionally, binding interactions between polypeptides and NCA at the ring N-H protons were also observed from the STD NMR spectra of BDG-NCA/poly(γ -ethyl-L-glutamate) and EG₂-Lys-NCA/PBLG (Supplementary Fig. 5).”

As a minor comment, authors should discuss why they do not observe pyroglutamate side reactions (intramolecular cyclization of the amino end-group with the adjacent benzyl ester) at room temperature in DCM (see the reference 12 given by authors). If ROP in DCM was performed at room temperature, was the MALDI-TOF presented as supplementary in figure 5 showing this pyroglutamate?

We thank the Reviewer for this comment. Polymerization in DCM is fast compared with that in DMF, which minimizes the pyroglutamate formation during such short time of polymerization. The end-group analysis revealed 99% was remained after the polymerization in DCM after 2 h ($[M]_0 = 0.4$ M, $[M]_0/[I]_0 = 100$). This is consistent with the previous results from Doty and co-workers, who estimated the formation of pyroglutamate to be ~1% per hour (*J. Am. Chem. Soc.*, **1957**, 79, 3961). The following discussion was added to the main text to highlight the benefits of accelerated polymerization on minimizing pyroglutamate formation (Page 7, highlighted):

“The accelerated polymerization kinetics in DCM help outpace side reactions during NCA polymerization. After the 2-h polymerization in DCM, the end-group analysis showed less than 1% loss of terminal amine groups, indicating negligible chain termination.”

Supplementary Fig. 5 in the original SI (Supplementary Fig. 7 in the revised SI) is the MALDI-TOF mass spectrometry showing the end-group analysis of PBLG 30-mer used for STD-NMR studies. The PBLG 30-mer was prepared in DMF following Ref 12 (*Polym Chem*, **2010**, 1, 514), and the synthesis was conducted at 4 °C to minimize the generation of pyroglutamate.

REVIEWERS' COMMENTS:

Reviewer #1 (Remarks to the Author):

I have carefully reexamined the revised manuscript and happy to see a significantly improved version. The authors have taken serious and substantial actions to address both reviewers' comments. The new manuscript is now publishable in Nat Commun from my point of view.

Reviewer #2 (Remarks to the Author):

After a detailed analysis of revised manuscript and all the novel data provided, I believe this paper can now be accepted in the present form in Nature Comm. I'd like to congratulate the authors for this very nice piece of work that will certainly impact a broad research area in polymer chemistry.